Effects of starch sugar by-product on rumen in vitro digestibility, in situ disappearance rate, and milking productivity of the lactating dairy cow

Choi Yongjun 1
Kim Suhun 1
Lee Sangrak 1
Na Youngjun ruminoreticulum@gmail.com 1 2
1 Department of Animal Science and Technology, Konkuk University , Seoul , South Korea
2 Animal Data Lab., Antller Inc. , Seoul , South Korea
Uversky Vladimir
Electronic publication date: 2022 Feb 22
Publication date: 2022
Volume: 10
Electronic Location ID: e12998
Received 2021 Oct 20; Accepted 2022 Feb 2
Copyright: ©2022 Choi et al.
Copyright year: 2022
Copyright holder: Choi et al.
License: This is an open access article distributed under the terms of the Creative Commons Attribution License, which permits unrestricted use, distribution, reproduction and adaptation in any medium and for any purpose provided that it is properly attributed. For attribution, the original author(s), title, publication source (PeerJ) and either DOI or URL of the article must be cited.
License URL: https://creativecommons.org/licenses/by/4.0/

Keywords: Feed value, Starch sugar by-product, Rumen in vitro digestibility, In situ disappearance rate, Lactating dairy cow

Funding: Ministry of Agriculture, Food and Rural Affairs (MAFRA) 121035021HD020 This work was supported by Korea Institute of Planning and Evaluation for Technology in Food, Agriculture and Forestry (IPET) through Livestock Industrialization Technology Development Program, funded by Ministry of Agriculture, Food and Rural Affairs (MAFRA)(121035021HD020). The funders had no role in study design, data collection and analysis, decision to publish, or preparation of the manuscript.

==============================
Objective

The purpose of the present study was to determine the effects of starch sugar by-product (SSB) feeding on the rumen in-vitro digestibility, in situ disappearance rate, and lactating dairy cow.

Methods

To determine the rumen in vitro digestibility, 50 mL of the buffer-rumen fluid mixture was dispensed into a 125 mL serum bottle containing 0.5 g of dry matter (DM) of substrates. Nitrogen gas (N2, 99.9% pure) was flushed into the serum bottles and three replications were incubated at 0, 2, 4, 8, 16, 24, and 48 h. To determine the in-situ disappearance rate, SSB was incubated for 0, 2, 4, 8, 16, 24, and 48 hours in nylon bags (5 × 10 cm, 45*m pore size) placed within the ventral sac of two cannulated Holstein cows.. A total of sixteen Holstein Friesian cows (60.5 ± 20.4 months old, 706.8 ± 3.4 kg initial body wieght) fed experimental diets during the experimental periods. The treatments were basal diet (control) and 3.0% DM of SSB, with the diet formulated according to national research council (NRC) nutrient requirements of dairy cattle guideline. An experiment was conducted with a randomized block design for six weeks based on body weight.

Results

Soluble fraction (fraction a) of DM and crude protein (CP) was 28.99 and 11.92%DM, fraction b of DM and CP was 44.63 and 31.61% DM, and c value of DM and CP was 26.38 and 56.47%DM. As an increase SSB level in total mixed ration (TMR), there was a decrease in gas production at 0, 16, and 48 h (p < 0.05). As an increase SSB level in TMR, there was a decrease in acetate to propionate ratio at 8, 16, 24, and 48 h (p < 0.05). Dry matter intake, milk production, and milk composition did not differ between the treatments. All blood profile contents did not differ between treatments.

Conclusion

A diet containing 3.0% SSB could be fed to ruminants without adverse effects on rumen fermentation.

Introduction

Recycling by-products could minimize environmental problems and help reduce livestock production costs (Bocquier & González-García, 2010). Refined starch is generally obtained from grains and it has been processed as various saccharides (modified starch) with glucose, oligosaccharide, or syrup (Gaffney, 2008). Since the starch industry is growing in recent years, more waste must be processed. Accordingly, it is necessary to conduct research on the recycling of starch sugar by-products.

The starch sugar by-product (SSB) is produced by the following process: (1) liquid starch passes through the filter to absorb useful saccharides into the center and (2) residue remains on the surface, which is separated using the knife (Kato, 1993). Sugar contents is present in the residue following the extraction of starch sugar from the filter (James, 2008). Furthermore, since the filter is constructed of silica (SiO2), SSB does not only contain sugar but also some silica, which is mixed during the cutting process of the filter (Rabelo, Da Costa & Vaz Rossel, 2015). Although many types of studies have been performed regarding the effect of starch in ruminants (Huntington, 1997; Mills, France & Dijkstra, 1999; Ortega Cerrilla & Mendoza Martínez, 2003; Harmon, Yamka & Elam, 2004; Svihus, Uhlen & Harstad, 2005), the studies of silica have not been as many as those of starch. Silica is known to be involved in the early stages of bone formation in animals, there was reported that it has effects on decreased cell wall digestibility of forage feed in ruminants (Soest, 1994). As a food additive, silicon dioxide is widely used, and no adverse effects have been reported in animals (Younes et al., 2018). Despite the fact that silica has been fed without negative effects in ruminants through forage and additives, it is important to determine whether it has negative effects on nutritional aspects when SSB is used as a feed ingredient in ruminants. Furthermore, studies on starch sugar by-products are rare in ruminants.

Therefore, the objectives of the present study were to analyze the physicochemical characteristics of SSB and to determine the effect of SSB level on rumen in vitro digestibility and in situ disappearance rate. In addition, we determined the effect of the SSB on the milk productivity of dairy cows.

Materials & Methods

Starch sugar by-product

The SSB used in the experiment originated from the starch sugar factory of Daesang Industry (Fig. 1; 884, Oehang-ro, Gunsan-si, Republic of Korea). As the physical properties can be affected by variations in the production process, the SSB samples were collected 10 times for 5 weeks and stored at −20 °C until the experiment was conducted. The distribution, variation and chemical composition, fatty acid composition, and amino acid composition of SSB were shown in Table 1, A1 and A2, respectively.

Figure 1 Starch sugar by-product (SSB) producing process.

Table 1 Physical and chemical characteristics of starch sugar by-product.

Items	Mean	Median	SD	MIN	MAX	Skewnessa	2 ×SESb	Kurtosisc	2 ×SEkd	
Physical characteristics										
Complex viscosity (hPa/s)	133.98	118.39	50.29	75.36	212.35	0.29	1.55	−1.48	3.10	
Particle, size (µm)	557	570	128	289	703	−1.16	1.55	1.07	3.10	
Density (DM g/cmc)	1.31	1.26	0.14	1.19	1.61	1.15	1.55	0.71	3.10	
Chemical compositionse										
DM (%)	61.43	61.80	3.52	55.20	65.83	−0.44	1.55	−0.79	3.10	
CP (%DM)	14.43	13.00	7.63	6.01	33.46	1.93	1.55	4.57	3.10	
EE (%DM)	23.26	21.40	7.41	10.66	36.99	0.22	1.55	0.42	3.10	
NDF (%DM)	35.67	34.10	11.28	13.15	51.22	−0.76	1.55	0.66	3.10	
ADF (%DM)	31.29	28.93	10.93	11.76	47.11	−0.20	1.55	−0.12	3.10	
Ash (%DM)	32.60	33.68	13.67	11.01	57.32	0.20	1.55	−0.06	3.10	
SiO2 (%DM)	30.54	31.55	12.81	10.31	53.70	0.18	1.55	−0.05	3.10	
WSC (%DM)	9.96	9.22	3.98	1.71	15.78	−0.48	1.55	1.37	3.10	
GE (Kcal/kg)	4,198	3965	1,033	2,341	6,005	0.11	1.55	0.48	3.10	
pH	4.14	3.71	0.80	3.41	5.33	0.97	1.55	−1.22	3.10	
Notes.

DM dry matter

CP crude protein

EE ether extract

NDF neutral detergent fiber

ADF acid detergent fiber

WSC water soluble carbohydrate

GE gross energy

SD Standard deviation

MIN Minimum value in database

MAX Maximum value in database

a The degree of asymmetry of a distribution around its mean where 0 ± 2 × Ses = normal.

b SEs, square root (6/n).

c Characterizes the relative peakedness or flatness of a distribution, where 0 ± 2 × Sek = normal.

d SEk, square root (24/n).

e ADF and SiO2 were analyzed to be residues left over following the analysis of NDF and ash, respectively.

Rumen in vitro digestibility and in situ disappearance rate

The experiment was carried out in the experimental farm at Pyeongchang-gun, Gangwon-do, South Korea (latitude 37.54° and longitude 128.44°). Two ruminally cannulated Holstein Friesian cows were assigned for rumen fluid collection and fed commercial concentrate pellet (dry matter [DM], 92.3%; crude protein [CP], 14.5% of DM; neutral detergent fiber [NDF], 31.8% of DM; acid detergent fiber [ADF], 11.6% of DM; ether extract [EE], 3.7% of DM; ash, 7.5% of DM) and rice straw (DM, 92.1%; CP, 5.5% of DM; NDF, 62.1% of DM; ADF, 36.6% of DM; EE, 1.4% of DM; ash, 5.2% of DM) ad libitum during the experiment. In the rumen in vitro digestibility test, McDougall’s buffer (McDougall, 1948), which is continuously purged with CO2 at 39 °C before usage, was mixed with rumen fluid at a 4:1 ratio (v/v). The total mixed ration (TMR) with SSB (0, 2, 4, 6, and 8% DM) samples were dried and milled to pass through a 1-mm screen (Wiley Mill; Thomas Scientific, Swedesboro, NJ, USA) and chemical composition showed in Table 2. The 50 mL of buffer-rumen fluid mixture was randomly dispensed into a 125 mL serum bottle filled with 0.5 g DM of substrates and the experimental unit was each serum bottle. N2 gas (99.9% pure) was flushed into headspace of the serum bottles and three replications were incubated at 0, 2, 4, 8, 16, 24, and 48 h.

Table 2 Ingredients and nutritional composition of different level of starch sugar by-product substrate on ruminal in vitro digestibility.

	SSB level (%DM)	
Items	0	2	4	6	8	
Ingredients (%DM)	
Timothy	30.00	30.00	30.00	30.00	30.00	
Beet pulp	10.00	10.00	10.00	10.00	10.00	
Corn	40.00	38.20	36.40	34.60	32.80	
Starch sugar by-product	0.00	2.00	4.00	6.00	8.00	
Soybean meal	20.00	19.80	19.60	19.40	19.20	
Chemical composition	
CP (%DM)	16.82	16.82	16.82	16.82	16.82	
EE (%DM)	2.78	3.15	3.52	3.89	4.26	
NDF (%DM)	55.72	55.45	55.18	54.92	54.65	
ADF (%DM)	25.61	26.08	26.56	27.03	27.51	
Ash (%DM)	4.99	5.48	5.98	6.48	6.97	
GE (kcal/kg)	4,438	4428	4418	4408	4399	
Notes.

SSB starch sugar by-product

CP crude protein

EE ether extract

NDF neutral detergent fiber

ADF acid detergent fiber

GE gross energy

Two ruminally cannulated Holstein cows were assigned for in situ disappearance rate measurements test. Dry matter, CP, EE, NDF, ADF, ash, and water-soluble carbohydrate (WSC) contents of SSB were 61.4, 14.4, 23.26, 35.7, 31.3, 32.6, and 9.96% of DM, respectively. The dried SSB samples were ground and screened using a 1-mm sieve prior to use in the experiment (Thomas Scientific, NJ, Swedesboro, NJ, USA). A total of 21 nylon bags (5 × 10 cm, 45 µm pore size; R510; ANKOM Inc.,NY, USA) containing 5 g of SSB have been placed in each rumen ventral sacs of two cannulated Holstein cows. These bags were incubated for 0, 2, 4, 8, 16, 24, and 48 h according to national research council (NRC) nutrient requirements of dairy cattle guidelines (NRC, 2001) in three replicate. After incubation, the nylon bags were washed in tap water, dried at 60 °C for 48 h, and then weighed for DM and CP analysis. Disappearance rate was assessed using the formula of Orskov (Menke et al., 1979) follow described in Choi et al. (2021). The following formula: P=a+b1−e−ct,

where P is the actual degradation after time t; a is the intercept of the degradation curve at time zero; b is the potential degradability of the component of the protein which will, in time, be degraded; c is the rate constant for the degradation of b; and t is time. Regression analysis was performed using SAS PROC REG (Version 9.4; SAS Institute Inc., NC, USA) for estimation of the fraction b.

The effective degradability (ED) of DM and CP was calculated using the following equation described in Choi et al. (2021): ED=a+b×c/c+k,

where k is the estimated rate of outflow from the rumen and a, b, and c are the same parameters as described above. The ED was estimated as ED2, ED5, and ED8 assuming rumen solid outflow rates of 0.02, 0.05, and 0.08/h, which are representative of low, medium, and high feeding intake, respectively.

Animal study method and experimental design

Animal research protocols were approved by Konkuk University Animal Care and Use Committee (approval number: KU16139).

A total of sixteen Holstein Friesian cows (60.5 ± 20.4 months old, 706.8 ± 3.4 kg initial BW) fed experimental diets (Table 3) during the entire experimental periods. Experimental animals were the commercial breed and obtained from Konkuk University experimental farm (latitude, 37.06° and longitude,127.86°). The average temperature and relative humidity were 25.3 ± 4.2 °C and 45.6 ± 11.57% during the entire experimental periods, respectively. Cows were allocated according to milk yield, parity, and days in milk and then allotted into six sawdust-bedded pens (two or three head/pen; 5 m × 10 m) with an individual feeding gate. The treatments were basal diet (control) and 3.0% DM of SSB (experimental), with the diet formulated according to NRC (Table 3). The feeding trial was conducted as a randomized complete block design for six weeks, which included two weeks of individual feeding gate adaptation, four weeks of adaptation to the experimental diet, and two weeks for data collection. Experimental diets were fed twice a day at 0800 and 1600 h in form of a TMR. Experimental diets, water, and mineral block were fed ad-libitum. The experimental animals were not euthanized after the end of the experiment, they have continuously raised after moving to a commercial farm as healthy.

Table 3 Formulation and chemical composition of control and starch sugar by-product diet in dairy lactating cows.

Items	Control	SSB	
Ingredient (%)			
Commercial mixed feeda	21.31	21.31	
Molasses	2.16	2.16	
Corn, flacked	4.26	4.26	
Corn gluten feed	1.99	1.99	
Wheat bran	1.99	1.99	
Rice bran	4.97	1.39	
SSBP	–	4.00	
Beetpulp pellet	3.41	4.55	
Cotton seed	4.97	4.97	
Browers grain, wet	9.23	9.23	
Alfalfa hay	8.10	8.10	
Oat hay	4.26	4.26	
Timothy hay	7.39	7.39	
Bermuda grass hay	2.84	2.84	
Klein grass hay	2.84	2.84	
Elvan	0.26	–	
Water	20.02	18.72	
Total	100.00	100.00	
Chemical composition			
DM (%)	64.99	64.96	
Forage ratio (%DM)	39.12	39.14	
NELb (Mcal/kg DM)	1.69	1.69	
TDNb (%DM)	73.87	73.87	
CP (%DM)	16.57	16.34	
EE (%DM)	5.38	5.38	
NFC (%DM)	29.61	29.44	
CF (%DM)	17.55	17.45	
NDF (%DM)	40.86	40.98	
ADF (%DM)	23.78	23.78	
Ash (%DM)	7.90	7.90	
Notes.

SSB starch sugar by-product

DM dry matter

NEL net energy for lactation

TDN total digestible nutrient

CP crude protein

EE ether extract

NFC non-fibrous carbohydrate

NDF neutral detergent fiber

ADF acid detergent fiber

a Commercial mixed feed formula, Corn grain 30.0%; Molasses 5.0%; Soybean meal 22.2%; Rapeseed meal 7%; Corn gluten feed 10.0%; Copra meal 5.6%; Parm oil meal 15.0%; Limestone 2.3%; Salt 0.8%; Sodium bicarbonate 0.8%; By-pass fat 0.3%; Vit & Mineral premix 1.0%.

b NEL and TDN was calculated by NRC (2001) model.

Physical and chemical analysis

Complex viscosity was analyzed using a rotational rheometer (DHR 1; TA instrument Ltd., DE, USA) at conditions of 0.1 to 100 Hz frequency and 20 °C. The particle size was determined by laser diffraction and scattering using particle size analyzer (LS 13-320; Beckman Coulter, Brea, CA, USA). Density of dried sample was analyzed using gas pycnometer (AccuPyc II 1340; Micromeritic Instrument Corporation, GA, USA).

All samples were dried in a drying oven (HB-503-LF; Hanbaek scientific technology, Buchun-si, Republic of Korea) at 60 °C for 48 h. Dry matter (DM; method No. 937.01), crude protein (CP; method No. 990.03), ether extract (EE, method No. 920.39), ash and silica (method No. 920.08) were analyzed according to AOAC method (AOAC, 2005). Neutral detergent fiber (NDF; method No. 2002.04) and acid detergent fiber (ADF; method No. 973.18) were analyzed with ANKOM Fiber Analyzer (A200; Ankom Inc., NY, USA) according to method of (Van Soest, Robertson & Lewis, 1991). Water soluble carbohydrate (WSC) was extracted using method of Kerepesi and Boross (Kerepesi, Toth & Boross, 1996) and was analyzed using phenol sulfuric acid method (Nielsen, 2017). Gross energy (GE) was determined using automatic bomb calorimeter (Parr 1261 bomb calorimeter; Parr Instruments Co., Moline, IL, USA). Gas production was measured with a 50 mL glass syringe (Hypodermic Glass Syringe; DHS Medical Co., Seoul, Republic of Korea). The pH values were measured using pH meter (Orion Dualster-F, Thermo fisher scientific, NJ, USA). Ammonia nitrogen was determined as previously described in Choi et al. (2019) according to a method of Chaney & Marbach (1962). The volatile fatty acid (VFA) was identified as previously described in Choi et al. (2019) using gas chromatography (HP 6890; Agilent Technologies, Santa Clara, CA, USA) equipped with an Omega Wax Fused Silica Capillary column (Length, 30 m 0.3 × 2 mm Df, 0.25 µm; Sigma-Aldrich Co, St. Louis, MO, USA) and flame ionization detector. The carrier gas was used He gas in gas chromatography.

Fatty acid composition analysis

For fatty acid (FA) analysis, SSB samples were extracted using a chloroform to methanol (2:1, v/v) solvent (Floch, 1957) and then methylated (Lepage & Roy, 1986). Methylated supernatant was injected into a gas chromatograph (Agilent 6890, NY, USA) equipped with a flame ionization detector and a capillary column (30 m × 0.25 mm × 0.25 µm; No. 122-3232; Agilent, Santa Clara, CA, USA) operated at 50 °C in the oven (Garces & Mancha, 1993). The inlet and detector temperatures were 180 and 250 °C, respectively. Helium was used as a carrier gas.

Amino acid composition analysis

To determine concentrations of amino acids (AAs) in SSB, each sample was placed in a volumetric flask to which was added 30 ml of 6N HCl. Then, the flask was hydrolyzed at 130 °C for 24 h min. The extracts were then passed through a 0.45 µm filter. For subsequent analysis, HPLC (Ultimate 3000; Thermo Fisher Scientific Inc., Waltham, MA, USA) was used as described by method of Henderson et al. (2000).

Milk yield, milk composition, and blood profiles

Milk yield was collected previously described in Choi et al. (2019) using a tandem milking system (Milking Parlor Auto Tandem; GEA Co., Düsseldorf, Germany) twice a day at 0300 and 1500 during the entire experimental period. Milk samples were collected in 20 mL tubes using the sampling port of a milking machine every week and stored at 4 °C. Before milk sampling, Anti-corrosive agents (Broad spectrum micro tabs II; Advanced Instrument Inc., Norwood, MA, USA) were added to prevent any chemical changes until the analysis of the milk composition. The milk composition was evaluated previously described in Choi et al. (2019) using near-infrared spectrophotometer (Milko-scan FT 6000; Foss electric Co., Hilleroed, Denmark).

Blood samples were collected at d 28 and d 42 after the end of the adaptation period. Blood samples were collected as previously described in Choi et al. (2021) via the jugular vein using 18-gauge needles and transferred to silicon-coated serum tubes (15 mL Vacutainer; BD, Franklin Lakes, NJ, USA). The serum and plasma were obtained by centrifugation at 1,000 * g at 4 °C for 15 min. Serum was stored at −70 °C until analysis and chemical compositions of serum were analyzed using an chemical analyzer (Model 7180 Clinical Analyzer; Hitachi Ltd, Tokyo, Japan) following the manufacturer’s manual. Reagents were purchased from commercial products (JW Medical, Seoul, Korea) to determine glucose, blood urea nitrogen (BUN), glutamic oxaloacetic transaminase (GOT), and glutamic pyruvate transaminase (GPT), and γ-glutamyltransferase (GGT). White blood cell, red blood cell, hematocrit, hemoglobin, and platelet were determined using hematology analyzer (VetScan HM2, Abaxix Inc., Holliston, MA, USA).

Statistical analysis

Data were analyzed using a MIXED procedure of SAS package program (SAS Inst. Inc., Cary, NC, USA) as a randomized completely block design. The model was, Yi(t)=μ+Ti+Ei(t),

where µ is average value, Ti is treatment value and Ei(t) is the error value. The fixed effect SSB concentration, and random effects were not considered. Polynomial orthogonal contrasts were used to determine SSB supplementation effect using the CONTRAST option. The crossing point of quadratic broken-line and the quadratic line was determined using NLIN code in order to determine proper SSB concentration in feed. Pairwise comparison was performed to determine SSB supplementation effect using the TTEST option. Outlier was excluded using the method of interquartile range (IQR). Least squares mean between treatments were assessed using a pairwise comparison method. Statistical difference and tendency were accepted at p-value less than 0.05 and 0.10, respectively. All means are presented as least square means.

Results

Dry matter and CP degradation parameters, and the ED values of SSB are presented in Table 4. Soluble fraction a of DM and CP content was 28.99 and 11.92% of DM, fraction b of DM and CP content was 44.63 and 31.61% of DM, and c value DM and CP content was 26.38 and 56.47 h−1. The ED2, ED5, and ED8 of DM content were 70.49, 66.54, and 63.28%, respectively and those of CP content were 42.45, 40.96, and 39.61%, respectively. Gas production, pH, and ammonia nitrogen content by SSB level are presented in Table 5. As increasing incubation time, gas production was increased in all treatment groups. At 0, 16, and 48 h, the result of gas production showed a significant difference among the treatments (p < 0.05), and as the amount of SSB in the TMR increased, the amount of gas production was quadratically reduced at 48 h (p < 0.05). As increasing incubation time, pH value was decreased in all treatment groups. The pH value showed a significant difference at 8 and 48 h during the rumen in vitro digestibility (p < 0.05). As increasing incubation time, ammonia nitrogen value was increased in all treatment groups. The ammonia nitrogen of SSB 6 and 8% treatment was lower than other treatment at 2, 4, 16, 24 (p < 0.05). Total VFA, acetate, propionate, and acetate/propionate (A/P) ratio by SSB level are presented in Table 6. As increasing incubation time, total VFA, acetate, and propionate content were increased in all treatment groups. Total VFA, acetate, and propionate did not differ among the treatments. As increasing incubation time, A/P ratio was decreased in all treatment groups. As the amount of SSB in TMR increased, there was a decrease in acetate/propionate ratio at 8, 16, 24, and 48 h (p < 0.05). Dry matter intake, milk production, and milk composition by SSB supplementation are presented in Table 7. Dry matter intake, milk production, and milk composition did not differ between the treatments. Blood profiles by SSB supplementation are presented in Table 8. All blood profile contents did not differ between treatments.

Table 4 Changes of in situ dry matter and crude protein disappearance rate of starch sugar by-product and corn in the rumen.

Item	In situ disappearance rate	
	DM	CP	
Incubation time (h)	
0	32.37 ± 4.49	13.87 ± 0.24	
2	49.12 ± 4.64	19.98 ± 1.36	
4	54.20 ± 4.89	22.64 ± 3.66	
8	56.10 ± 6.30	23.09 ± 0.07	
16	57.11 ± 3.24	26.12 ± 0.26	
24	63.16 ± 4.53	28.13 ± 1.90	
48	73.62 ± 4.13	43.53 ± 0.52	
Degradation parametera	
a (%DM)	28.99	11.92	
b (%DM)	44.63	31.61	
c, h−1	26.38	56.47	
EDb (%)	
ED2	70.49	42.45	
ED5	66.54	40.96	
ED8	63.28	39.61	
Notes.

a a, water soluble fraction which is rapidly washed out of bags and assumed to be completely degradable; b, the slowly degradable fraction; c, the rate of degradation per hour.

b ED, effective degradability; A fractional rate of passage out of the rumen, which was assumed as 0.02, 0.05 and 0.08/h.

Table 5 Effect of starch sugar by-product on ruminal in vitro gas production, pH, and ammonia nitrogen.

	SSB (%DM)			
Incubation time (h)	0	2	4	6	8	SEM	p-value	
	Gas production, ml			
0	0.00	0.00	0.00	0.00	0.00	–	–	
2	13.00	12.33	11.33	10.67	9.00	1.10	0.17	
4	20.33	20.00	21.33	20.00	17.33	1.10	0.16	
8	36.00a	31.67b	29.67bc	26.33cd	22.33d	0.87	<0.01	
16	64.67ab	67.33a	56.00ab	64.00b	52.00b	2.86	0.01	
24	69.67	65.00	68.67	67.33	63.33	2.86	0.27	
48e	96.67a	97.33a	79.33b	81.33b	81.67b	1.50	<0.01	
	pH			
0	6.90	6.90	6.90	6.90	6.90	–	–	
2	6.99	7.06	7.03	7.00	7.02	0.03	0.49	
4	6.96	6.97	7.00	6.97	6.91	0.03	0.29	
8	6.76ab	6.83a	6.75b	6.80ab	6.78ab	0.01	0.03	
16	6.59	6.60	6.64	6.65	6.63	0.02	0.08	
24	6.55	6.52	6.58	6.56	6.55	0.02	0.25	
48	6.50a	6.44ab	6.47ab	6.42a	6.45ab	0.01	0.02	
	Ammonia nitrogen (mg/100 ml)			
0	0.50	0.50	0.50	0.50	0.50	–	–	
2	1.11a	1.27a	1.27a	0.73b	0.59b	0.04	<0.01	
4	0.62a	0.54ab	0.54ab	0.40b	0.40b	0.05	0.03	
8	0.82	0.59	0.48	0.48	0.43	0.10	0.14	
16	0.93a	0.83a	0.83a	0.63b	0.66a	0.06	0.03	
24	4.09a	4.03a	3.45a	2.01b	2.15b	0.18	<0.01	
48	5.44	5.38	5.22	4.94	5.25	0.29	0.77	
Notes.

SSB starch sugar by-product

SEM standard error of the mean

abcd Means in the same row with different superscrips differ significantly (P < 0.05).

e It had sigmoidally decreased as an increase SSB level (Equation, Y=80.78+16.221+x3.7734.8910.53, R2 = 0.989).

Table 6 Effect of starch sugar by-product on ruminal in vitro volatile fatty acid synthesis.

	SSB (%DM)			
Incubation time (h)	0	2	4	6	8	SEM1	p-value	
	Total VFA (mM)			
0	22.26	22.26	22.26	22.26	22.26	2.03	–	
2	22.47	23.00	22.43	23.07	24.05	0.63	0.41	
4	29.89	30.94	30.63	31.89	30.52	1.53	0.92	
8	41.02	41.61	43.67	41.80	40.88	1.21	0.53	
16	61.69	62.08	60.84	57.56	57.28	2.14	0.39	
24	66.85	69.76	67.46	68.18	67.18	3.35	0.97	
48	82.31	81.07	87.81	84.13	86.28	2.58	0.39	
	Acetate (mM)			
0	13.21	13.21	13.21	13.21	13.21	0.99	–	
2	13.38	13.67	13.22	13.71	14.25	0.39	0.44	
4	17.64	18.16	17.91	18.69	17.73	0.81	0.89	
8	23.38	23.74	24.62	23.30	22.74	066	0.41	
16	33.65	33.29	32.21	30.53	29.98	1.10	0.14	
24	35.64	36.22	35.60	35.48	34.56	1.73	0.97	
48	43.13	41.98	45.02	43.15	43.59	1.30	0.60	
	Propionate (mM)			
0	5.47	5.47	5.47	5.47	5.47	0.49	–	
2	5.72	5.79	5.64	5.84	6.13	0.18	0.40	
4	7.69	7.99	7.85	8.32	8.04	0.54	0.94	
8	11.34	11.43	12.63	12.04	12.06	0.45	0.31	
16	18.60	18.58	19.07	18.22	18.76	0.79	0.96	
24	20.57	22.32	21.41	22.44	22.80	1.26	0.72	
48	25.12	25.30	28.24	27.91	29.02	1.01	0.06	
	A/P ratio			
0	2.42	2.42	2.42	2.42	2.42	0.04	–	
2	2.34	2.36	2.34	2.35	2.33	0.02	0.72	
4	2.30	2.28	2.28	2.27	2.21	0.05	0.75	
8	2.06ab	2.08a	1.95bc	1.94c	1.89c	0.02	<0.01	
16	1.81a	1.79a	1.70ab	1.68ab	1.60b	0.04	0.01	
24	1.73a	1.63abc	1.66ab	1.58bc	1.52c	0.03	0.01	
48	1.72a	1.66ab	1.60ab	1.54ab	1.50b	0.04	0.02	
Notes.

SSB starch sugar by-product

SEM standard error of the mean

VFA volatile fatty acid

A/P ratio acetate to propionate ratio

abc Means in the same row with different superscrips differ significantly (P < 0.05).

Table 7 Dry matter intake, milk production and composition of dairy lactating cows fed the control and starch sugar by-product diet.

Items	Control	SSBa	SEM	P-value	
Dry matter intake (kg/cows/day)	27.47	24.52	0.47	0.45	
Milk production					
Milk yield (kg/cow/day)	29.01	30.02	0.94	0.46	
4%FCMb (kg/cow/day)	29.45	32.06	1.14	0.14	
FPCMc (kg/cow/day)	29.45	31.47	1.06	0.47	
Milk composition					
Fat (%)	4.13	4.45	0.18	0.21	
Protein (%)	3.36	3.25	0.06	0.22	
Lactose (%)	4.64	4.78	0.05	0.06	
Solid not fat (%)	8.79	8.73	0.06	0.47	
Milk urea nitrogen (ng/ml)	11.20	11.78	0.49	0.36	
Somatic cell counts (10ccell/ml)	273.67	192.75	115.23	0.39	
Notes.

SSB starch sugar by-product

SEM Standard error of the mean

a Replacement of 3.0% DM of SSB in total mixed ration.

b 4% Fat corrected milk (4%FCM) was calculated from 4%FCM = 0. 4 × milk yield + 15 × milk fat yield.

c Fat-protein corrected milk (FPCM) was calculated from FPCM = milk yield × (0.337 + 0. 116 × milk fat (%) + 0. 06 × milk protein (%).

Table 8 Blood profiles of lactating dairy cows fed the control and starch sugar by-product diet.

Items	Control	SSBa	SEM	P-value	
Glucose (mg/dL)	37.44	30.19	2.89	0.10	
BUN (mg/dL)	14.38	15.08	0.50	0.34	
Cholesterol (mg/dL)	296.13	291.25	25.71	0.90	
GOT (IU/L)	70.25	62.25	6.56	0.40	
GPT (IU/L)	26.88	26.19	1.34	0.72	
GGT (IU/L)	38.06	31.31	2.90	0.12	
WBC (103/µL)	12.03	12.33	1.61	0.90	
RBC (106/µL)	6.19	5.93	0.20	0.38	
Hematocrit (%)	33.86	33.01	1.03	0.57	
Hemoglobin (g/dL)	52.60	33.38	18.55	0.31	
Platelet (103/µL)	332.00	366.31	28.39	0.41	
Notes.

SSB starch sugar by-product

SEM standard error of the mean

BUN blood urea nitrogen

AST aspartate aminotransferase

ALT alanine transferase

GGT γ-glutamyltransferase

WBC white blood cell

RBC red blood cell

a Replacement of 3.0% DM of SSB in total mixed ration.

Discussion

In the starch sugar production process, SSB is separated by the physical method such as a sharp blade (Fig. 1) and it could have effects on the change of physical and chemical composition of SSB. The physical and chemical characteristics of collected SSB showed a high degree of variability in this experiment (see Table 1). It is considered to be one of the factors that make it difficult to use as a feed ingredient. The most ash content of SSB might be considered as silica content (SiO2) in this study (see Table 1), silica contents was reported that have a negative effect on ruminants such as a decrease of forage utilization (Soest, 1994). Thus, this may be considered a weakness in the nutritional aspects as a ruminant feed component. However, SSB has comparable gross energy and similar ED value within the rumen compared to corn (Silva et al., 2020). SSB is considered that have some potential as an energy-feed ingredient. Moreover, the content of CP in SSB ranged from 6.01 to 33.46% of DM, indicating that it could be utilized as a protein supplement (see Table 1). Soybean meal is mainly used as a protein source in livestock feed. Although ED value of CP in SSB (39.6 to 42.5%, Table 4) was lower than those of soybean meal (63 to 71%, (Heuzé, Tran & Kaushik, 2015)), it could be provided another option to choose protein ingredients in livestock feed. During the in vitro rumen digestibility trial, the amount of gas produced means those of organic digestibility by bacterial microbes (Theodorou et al., 1994). Gas production showed the lowest at 48 h in SSB levels of 4 to 8% groups compared to those of 0 and 2% groups (see Table 5). There was reported that an increase of lipid sources in feed has a negative effect on rumen fermentation (Johnson & Johnson, 1995). In the rumen in vitro digestibility trial, as a level of SSB in the experimental feed increased, EE and ash content was linearly increased (see Table 2). Therefore, the decrease of gas production in the rumen in vitro digestibility trial might be explained by an increase in both lipids and ash contents in the experimental feed. In addition, a level of 2% SSB based on the dry matter in feed is considered to be the highest level that can be fed to animals without affecting their digestibility. Despite a significant difference in pH level at 8 and 48 h, the pH content in the rumen was within a normal range (Dijkstra et al., 2012), thus the difference in pH alone hardly explains the negative effect. However, the significant difference in ammonia nitrogen content might be mean that inhibit temporarily microbial fermentation according to SSB level in the feed (Hristov & Ropp, 2003). Ammonia nitrogen of 8 and 10% SSB treatment were lower than those of other treatments until 24 h (see Table 5). Nevertheless, ammonia nitrogen content did not differ at 48 h among treatments, it might consider that as degradation of organic matter in feed by the fermentation, it mitigated toxicity in the rumen to microbes such as lipids (Jarvis & Moore, 2010). Totally, a high amount of SSB (about 8% over) supplementation considers that seriously inhibit rumen fermentation. In the result of VFA content, SSB supplementation seems that did not inhibit producing VFAs during rumen in vitro digestibility. Total VFA, acetate, and propionate did not differ among treatments. However, the A/P ratio was decreased as an increase in the SSB level of feed (see Table 6). It means that supplementation of SSB supplied non-fibrous carbohydrates in TMR (James, 2008). In other words, it considers that the SSB could be used as an energy source of feed. In addition, increasing the SSB level of feed means that increasing the ratio of NFC to fibrous material in the feed. It is possible to explain the decrease in the A/P ratio of this study by the fact that propionate is produced by microbes using NFC in the rumen. There was reported that the A/P ratio in rumen fermentation was a good indicator of milk fat synthesis and acidification (NRC, 2001). In this experiment, as decreasing A/P ratio in the rumen, it considers that a high level of SSB supplementation could increase milk yield but might affect rumen acidification and negative effect on feed utilization (Russell, 1998).

In the rumen in vitro digestibility trial, the negative effects of SSB supplementation did not significantly observe in the 2% supplementation group. When feeding 4.0% of DM SSB, the negative effects of SSB supplementation were significantly observed during rumen in vitro digestibility. When describing the biological action of microorganisms mathematically, the sigmoid function is generally used. The gas production was predicted as 96.96% using a fitted sigmoid curve when feeding 3.0% of DM SSB in the rumen in vitro digestibility test (see Table 5). Thus, we decided to feed 3.0% of DM SSB in the feeding trial. In the feeding trial, as fed TMR including 3.0% of DM of SSB, the performance did not show a significant difference in the DMI, milk production, and milk composition compared with those of control. In the in vitro trial, the A/P ratio was decreased as an increase SSB, which could mean that the digestibility of fibrous materials was lower than those of non-fibrous material in the SSB. The decrease of the A/P ratio in the rumen has an effect on an increasing milk yield of lactating cow (Sutton et al., 2003), the milk yield did not show a significant difference between treatments during the feeding trial. This result considers that the NDF and ADF fraction in the SSB seems to play a sufficient role as roughage in the rumen. Furthermore, in the milk productivity and blood profiles, it did not show a negative effect with fed 3.0% DM of SSB on the lactating cows. Although the SSB has disadvantages such as high silica contained ash content, high EE content, and low nutritional uniformity, it has enough proper CP content and good gross energy as a feed ingredient (see Table 1). High levels of silica and fat content in the SSB can reduce the risk by reducing the fed amount to the ruminant. Furthermore, about 12,000 tons of starch sugar by-products are dumped per year in South Korea (Park, Oh & Kim, 2017) if approximately 0.5 percent of the TMR will substitute as the SSB, all of them can be recycled. If the proper amount of SSB is used for animal feed ingredients, it could be diminished not only the environmental pollution but economic loss by saving the budget for disposal. In this study, 3.0% DM of SSB in TMR did not negatively affect the milk production of lactating cows, which makes it a reasonable suggestion for feeding levels.

Conclusions

The SSB levels of 3.0% of DM in the diet can be used for the lactating cow without adverse effects on milk productivity and blood profile. However, the SSB might have some negative effects on the ruminant when dietary levels of SSB were increased to 3.0% of DM due to high silica-contained ash and high EE content. On the other hand, the SSB has proper CP content and good gross energy as a feed ingredient. Therefore, the TMR contained 3.0% of DM of SSB can be used in the lactating cows as an energy and protein source without adverse effects.

Supplemental Information

Supplemental Information 1 Fatty acid composition of starch sugar by-product

Click here for additional data file.

Supplemental Information 2 Amino acid content of starch sugar by-product

Click here for additional data file.

Supplemental Information 3 Raw data

Raw material ingredients, feed formula, in situ disappearance, in vitro digestibility test , and animal test results.

Click here for additional data file.

Supplemental Information 4 Author Checklist

Click here for additional data file.

Additional Information and Declarations

Competing Interests

Author Contributions

Animal Ethics

Data Availability

All authors declare that they have no competing interests. Yougnjun Na served as Chief Executive Officer for Antller Inc.

Yongjun Choi conceived and designed the experiments, performed the experiments, analyzed the data, prepared figures and/or tables, and approved the final draft.

Suhun Kim conceived and designed the experiments, performed the experiments, prepared figures and/or tables, and approved the final draft.

Sangrak Lee and Youngjun Na analyzed the data, authored or reviewed drafts of the paper, and approved the final draft.

The following information was supplied relating to ethical approvals (i.e., approving body and any reference numbers):

All research protocols were approved by Konkuk University Animal Care and Use Committee

The following information was supplied regarding data availability:

The raw data is available in the Supplementary File.

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
