# Peer review of "Effects of starch sugar by-product on rumen in vitro digestibility, in situ disappearance rate, and milking productivity of the lactating dairy cow"

_PeerJ, doi:10.7717/peerj.12998_

## Round 0.1 · original submission · Major Revisions

Please address the concerns of both reviewers and revise the manuscript accordingly.

Reviewer 1 ·

Excellent Review

This review has been rated excellent by staff (in the top 15% of reviews)
EDITOR COMMENT
Thank you very much for providing your insightful and constructive critiques. I am sure that your comments will help the authors to greatly improve their manuscript.

Basic reporting

The presented manuscript “Effects of starch sugar by-product on rumen in vitro digestibility, in situ disappearance rate, and milking productivity of the lactating dairy cow” is directly related to several contemporary issues and thus provides a great value to the community. However, in its current form, the presented manuscript suffers from several shortcomings.

The English language and readability of the presented manuscript should be improved. Although being a non-native English speaker, this reviewer has identified several parts that are grammatically incorrect and also the overall fluency of the text could be improved. This reviewer suggests to ask an English-speaking colleague to proof-read the manuscript or pay for a professional service. A couple of examples with unclear phrasing or mistakes can be found in lines: 37, 46-48, 56, 62-64, 75-77, 84, 88, 94, 102, 208, 219-220, 241, 246-250, 256, 261-272 (but please, focus on the entire text).

Overall, the structure of the manuscript is appropriate. This reviewer would just suggest to extend the introductory and result sections and possibly reorganize the individual method sections as described below.

The introduction could be improved and extended to better justify the need for this study. In the introduction, only a handful of references were used. In lines 53-55, it is not clear whether this is the authors’ speculation or whether it is a known fact. Similarly, in the line 55, it is not clear where the assumption that silica contaminates SSB comes from. In line 58, more than just two studies could be cited. In line 59, the “rare” research studies should also be properly cited to better place the current study in the context of known literature. Line 62 was not researched so its importance in the introduction is not clear – or can the predisposition to urinary tract stones be implied from the blood urea nitrogen (if so, it should be specified in the text)? In general, it is not clear how the blood samples are relevant for the study so it could be introduced. Similarly, the impact of fatty acid synthesis on milk production and composition is not introduced.

Experimental design

The objectives of the study are within the scope of the journal. The research questions and aims of the study could be defined more clearly.

The end of the introduction (lines 61-65) implies that the goal of the study is to: a) investigate the impact of silica present in SSB on cows’ physiology; b) determine whether SSB-enriched feed can be used in practice. However, the effects resulting from SiO2 presence in SSB were not sufficiently studied. Firstly, the values for SiO2 are missing in the raw data table characterizing the chemical composition of SSB. A control sample indicating if there is SiO2 present in the standard feed is missing. It would be better to quantify SiO2 also in the Table 2 and Table 3 to indicate the differences between standard and SSB-enriched feed (this reviewer does not ask for more experiments at this stage of the project; however, clear discussion of these aspects should be provided). In lines 244-245, the authors suggest that the ash is made up of silica. In its current form, this is just a speculation so this reviewer suggests to tone it down. Moreover, the effect of SiO2 is not covered in the results section. Overall, it would probably be better to modify/remove the parts regarding SiO2 as it is neither well introduced nor well investigated. A possible suggestion is either to improve these parts or keep the SiO2 part only for the discussion.

The remaining parts investigating the in vitro digestibility and in vivo disappearance of SSB are performed and described well. Methods are described with sufficient detail and information to replicate (if the English is improved) and the methodology used is adequate to the study.

Validity of the findings

The authors provided processed as well as raw data and described well the statistical analyses. Several comments regarding the data provided are below.

One of the most important finding relies in the altered gas production, ammonia nitrogen and A/P ratio upon increased SSB levels. The authors attribute the decreased gas production to inhibited microbial fermentation due to increased lipid and silica content (lines 252-253 and 261-262). However, note that sugars and starches (which constitute the nature of SSB) are broken down easily and quickly. It seems that the gas production is directly linked to digestibility of the feed; where with good digestibility the energy produced is used to increase weight and production of milk, which is associated with decreased enteric CH4 production (doi.org/10.3390/ani11071870; doi:10.1017/S1751731113000876). In line with these observations, with increasing SSB levels the authors identified increased levels of propionate (and thus decreased A/P ratio) which is known to be an end-product of starch and sugar fermentation and whose presence is associated with decreased CH4 and CO2 levels (Moran, 2005, How the Rumen Works). Considering the fact that the milk production was slightly increased and there were no dramatic changes to the pH, couldn’t this hypothesis, the simple fact that the feed was enriched for starch, better explain the results of the study? How the authors can be sure that the SSB-enriched feed does not provide enough nutrients to grow cellulolytic bacteria (line 266)? This is quite an overstatement since the authors did not investigate the grow of bacteria.

Additional comments

Typical reader of scientific literature first reads the results section and only in the case when something needs to be validated he/she seeks the methods. In the presented manuscript, the results are written in a way that it is difficult to follow, partly due to a high number of shortcuts used. Please consider re-phrasing the results to make this section more readable. Additionally, results could be extended to cover all aspects of the manuscript and in greater detail. Additionally, there are shortcuts that should be explained earlier. For example, DM, TMR, NRC in the abstract are not explained.

The organization of tables is a bit confusing. First, the authors present a composition of feed used in the in vitro (Table 2) and in vivo assays (Table 3), respectively. Then, they present data in the reverse order, i.e. first the in situ disappearance rate (Table 4), followed by the in vitro gas production (Table 5) and fatty acid synthesis (Table 6). Wouldn’t it be better to organize the tables in a more logical way and place the tables summarizing composition and results of the underlying assays next to each other? This would also help to understand the correlations between the composition and results of the analysis. In fact, it would be very helpful to provide a correlation analysis between the composition and analysis results to indicate a potential relationship between the variables and observed physiological changes (such as between EE, ADF, Ash and gas production, pH, ammonia nitrogen, fatty acids). For example, ADF is a good indicator of digestibility (doi:10.3390/foods8090364 and references therein) so it would be nice to see it confirmed also by this study. Additionally, a graphical representation of the data (similar to those in the supplementary excel file) would be highly appreciated by readers (indicating also the distribution/variability of the data).

Table 1. There are multiple categories (CP, EE, NDF, ADF, Ash, SiO2, WSC) that are expressed as % DM. However, when these numbers are summed up, it gives > 177%. Similar discrepancy can also be found for the individual measurements in the raw data excel table “1-2. Chemical characteristscs” (check also spelling of the word “characteristics” in all the tabs of the raw data table). Is it because these categories overlap? It is basically explained by the lines 244-245 where the authors suggest that the ash is made up of silica. But maybe it could be briefly explained somewhere in the text not to confuse readers.

In raw data excel file tab “2-1. IN SITU disapperance - DM” (please, correct the typo), the time point 0 in the graph is based on value 32.63 even though the value should be 32.37 as demonstrated in the Table 4. Similar discrepancy is present for the time point 8.

The authors should indicate based on which criteria they decided to use 3% DM of SSB for their “feeding trial”.

The authors should discuss the short duration (only six weeks) of the study. It is reasonable to assume that life-long supplementation of SSB could uncover more dramatic effects on cows well-being/milk production/weight gain.

Reviewer 2 ·

Basic reporting

no comment

Experimental design

no comment

Validity of the findings

no comment

Additional comments

Dear authors,
Overall work here, I believe the authors have done an exemplary job in preparing this manuscript. The level of scientific rigor is apparent, and the attention to detail with regard to every aspect of the replication is appreciated.

I have a few minor suggestions that the authors might consider, but all of them would move forward.
1. The manuscript the authors discuss the connection between abstract statements and those of passive voice. This discussion includes some clarification from real data numbers in the world (statistical data in present), but I wonder if additional clarification about the connection between the abstract characteristics and passive voice would be worth discussing here.

2. On line 79 in the Materials section, the authors mention that the statements were evaluated so that
“the abstract and concrete versions used equally common language.” Was that equality evaluated in any specific way?
3. Please mention the different time frame variability and accuracy.

4. Please clearly mention all the reagent and chemical sources in this format; Company name, Manufacture location, Country in the manuscript.

6. The values are showing in table 8 is very close to human is this same in the dairy cow?

---

## Round 0.2 · accepted · Accept

All issues pointed out by the reviewers were adequately addressed and therefore I am pleased to accept your revised manuscript now.

Reviewer 1 ·

Basic reporting

The authors significantly improved the text quality in the revised version of the manuscript. Some language issues remained unresolved, thus this reviewer suggests one more proofreading round by an English speaking colleague.

Experimental design

The authors handled this reviewer’s comments well and applied the requested changes.

Validity of the findings

This reviewer suggests to rephrase the modified part and correct the English to improve its readability.

Reviewer 2 ·

Basic reporting

The manuscript has been modified and approved for no further comments.

Experimental design

Looks good

Validity of the findings

Excellent and much improved